# Maternal hybrid immunity and risk of infant COVID-19 hospitalizations: national case-control study in Israel

Joshua Guedalia [1,7], Michal Lipschuetz [2,3,4,7] ✉, Adva Cahen-Peretz[2], Sarah M. Cohen [2], Yishai Sompolinsky[2], Galit Shefer[5], Eli Melul[5], Zivanit Ergaz-Shaltiel[6], Debra Goldman-Wohl[2], Simcha Yagel[2], Ronit Calderon-Margalit[1,8] & Ofer Beharier [2,4,8] ✉

Hybrid immunity, acquired through vaccination followed or preceded by a COVID-19 infection, elicits robust antibody augmentation. We hypothesize that maternal hybrid immunity will provide greater infant protection than other forms of COVID-19 immunity in the first 6 months of life. We conducted a case-control study in Israel, enrolling 661 infants up to 6 months of age, hospitalized with COVID-19 (cases) and 59,460 age-matched non-hospitalized infants (controls) between August 24, 2021, and March 15, 2022. Infants were grouped by maternal immunity status at delivery: Naïve (never vaccinated or tested positive, reference group), Hybrid-immunity (vaccinated and tested positive), Natural-immunity (tested positive before or during the study period), Full-vaccination (two-shot regimen plus 1 booster), and Partial-vaccination (less than full three shot regimen). Applying Cox proportional hazards models to estimate the hazard ratios, which was then converted to percent vaccine effectiveness, and using the Naïve group as the reference, maternal hybrid-immunity provided the highest protection (84% [95% CI 75-90]), followed by full-vaccination (66% [95% CI 56-74]), natural-immunity (56% [95% CI 39-68]), and partial-vaccination (29% [95% CI 15-41]). Maternal hybrid-immunity was associated with a reduced risk of infant hospitalization for Covid-19, as compared to natural-immunity, regardless of exposure timing or sequence. These findings emphasize the benefits of vaccinating previously infected individuals during pregnancy to reduce COVID-19 hospitalizations in early infancy.

Infants are at increased risk for serious COVID-19 disease with hospitalization for acute respiratory failure as compared with older children[1–3]. In the US, hospitalization rates among infants aged <6 months were approximately five times higher during the peak week of Omicron predominance compared to the Delta predominance period[3]. While messenger RNA (mRNA) vaccines (BNT162b2-Pfizer and mRNA-1273 Moderna) were authorized for children as young as 6 months[3,4], prevention of infection and illness for younger infants remains an issue of significant concern.

Maternal anti-SARS-CoV-2 antibodies play a critical role in maternal antiviral immunity[5]. Moreover, maternal immunoglobulin G (IgG) antibodies cross the placental barrier, providing the first line of defense for neonatal humoral immunity. Previous studies have demonstrated the effectiveness of maternal immunization in

preventing pertussis and influenza infections in infants[6–8]. Moreover, a 2023 study demonstrated the use of RSVperF vaccine during pregnancy to limit infant illnesses[9]. We and others have previously described the significant role of a third maternal mRNA vaccine dose during pregnancy in reducing the risk for mother[10] and infant hospitalizations due to COVID-19[11]. These findings fit with previous reports by Halasa et al. that showed an association between maternal vaccination with the second dose of mRNA vaccine and reduced risks of COVID-19-related hospitalizations, and critical disease among infants younger than six months[12]. These data support the notion that anti-SARS-CoV-2 vaccine-mediated immunity has a significant clinical impact for both mother and offspring. Nevertheless, the role of vaccination following SARS-CoV-2 infection remains to be elucidated.

The dynamics of anti-SARS-CoV-2 antibody levels following infection during pregnancy were described by us and others[5,13]. Similar to what was described in the general population, we found waning of anti-SARS-CoV-2 antibody levels in those infected with SARS-CoV-2 either before or during pregnancy[13]. We further showed that hybrid immunity, i.e., immunity conferred by the combination of infection and vaccination, elicits higher levels of neutralizing antibodies in both maternal and umbilical cord blood, detected at delivery, when compared with recovered unimmunized pregnant women[13]. These results are consistent with data obtained in non-pregnant populations, showing that hybrid immunity reinforced protection against reinfection[14,15].

The clinical significance of maternal hybrid immunity in COVID-19 protection, both in mothers and their offspring, is unclear. Moreover, how hybrid immunity compares with natural immunity (i.e., immunity acquired by previous infection alone), remains an open fundamental question. Addressing these knowledge gaps is imperative, particularly considering the suboptimal vaccination response observed in previously infected pregnant women.

We therefore aimed to estimate the effectiveness of maternal hybrid immunity during pregnancy against COVID-19-related hospital admissions for infants in their first six months of life. To do so, we compared subgroups of infants categorized by maternal immunity-conferring events and analyzed real-world national-level data obtained in Israel during the periods of Delta and Omicron variant circulation.

## Results
### Participants
Characteristics of infant cases and controls are shown in Table 1, in addition to participants characteristics by maternal immunity group. Study participants included 661 infants that were admitted to hospital due to COVID-19 up to age of 180 days (case infants, age distribution of the cases, Supplementary Fig. 1), and 59,460 that were not hospitalized (control infants), (see Fig. 1- study flow chart and Supplementary fig. 2). Cases and controls were similar in most characteristics; however, primiparity was more common among the cases than controls (33.9% vs. 28.3%, respectively) and grandmultiparity was less common (11.6% vs. 15.3%, respectively). Mothers of cases had more tests for SARS-COV-2 than controls ≥5 documented tests: 46.0% vs. 36.9%, respectively). Cases and controls were grouped by mother's immunity status at the time of delivery: Naïve (reference group), Natural immunity, Hybrid immunity, Partial vaccination, or Full vaccination (3 doses).

Maternal characteristics were analyzed according to immunity status sub-groups. Mothers with hybrid immunity were older compared to mothers in the natural immunity and naïve groups, with 71.6% of them aged >27 years, as opposed to 64.1% and 60.5% in the latter groups, respectively. Additionally, they underwent SARS-CoV-2 testing more frequently, with 48.0% of them having undergone ≥5 tests, as compared to 40.5% and 20.3% in the natural immunity and naïve groups, respectively (Table 1).

A total of 20 of the 661 case infants (3.0%) were born to mothers with hybrid immunity, 46 (7.0%) to mothers with natural immunity, 95 (14.4%) to mothers with full vaccination, 272 (41.1%) partial vaccination, and 228 (34.5%) to naïve mothers. In comparison, the control group comprised 59,460 infants distributed as follows: 5832 (9.8%) hybrid immunity, 5430 (9.2%) natural immunity, 12,212 (20.5%) full vaccination, 21,287 (35.8%) partial vaccination, and 14,699 (24.7%) born to naïve mothers. Maternal post-partum vaccination could impact the risk of COVID-19 infection among infants[16,17]; we therefore analyzed the frequency of post-partum vaccination among all subgroups and found similar rates (ranging from 7 to 10%). The distribution of the study population by maternal immunity status for case and control infants, over the study period, is illustrated in Supplementary fig. 3.

### Clinical severity of COVID-19 hospitalization among cases
As shown in Table 1, of the 661 infants hospitalized due to COVID-19, 33 (5.0%) were admitted to the pediatric critical care units (PICU), with no significant difference in PICU admission rates among maternal immunization groups (ANOVA, $p = 0.852$). The overall median length of stay of COVID-19 hospitalization for all cases was 2 days (IQR 1–3)). There were no significant differences in mean length of stay between infants of mothers in the hybrid immunity group and all other groups (ANOVA, $2.30 \pm 1.3$ days, $p = 1.000$). However, infants born to mothers in the full vaccination group experienced a shorter median length of stay of 2 (IQR 1–2) days (Kruskal–Wallis, $p = 0.008$) than those born to mothers in the naïve group, with a mean of 2.43 ($\pm 1.8$) (ANOVA, $p = 0.009$, Bonferroni post-hoc correction). No deaths were recorded among the infants hospitalized for COVID-19.

### Maternal mediated immunity estimated effectiveness
Using Cox proportional hazards models to estimate the hazard ratios (HR), we compared the effectiveness of maternal hybrid, natural, and vaccine-induced (full and partial) immunity, with naïve mothers as reference group, against COVID-19-associated hospitalization of infants below six months of age. Our results demonstrated that maternal hybrid immunity conferred the greatest protection at 84% (95% CI: 75–90), followed by full vaccination at 66% (95% CI: 56–74), natural immunity at 56% (95% CI: 39–68), and partial vaccination at 29% (95% CI: 15–41; Fig. 2). Hybrid immunity effectiveness differed (84% vs. 66%, 56%, and 29%) significantly from all other groups ($p < 0.005$, $p < 0.001$, and $p < 0.001$, respectively). In a sensitivity analysis including infants who were considered non-eligible as cases (i.e., tested positive for COVID-19, but hospitalized for reasons unrelated to COVID-19), similar overall trends were observed (Supplementary Table 2).

We performed separate analyses of infant hospitalizations during the Delta and Omicron variant surges. During the Delta variant surge 173 infants were admitted, whereas 486 infants were admitted during the Omicron surge, a 2.8-fold increase. These cases were matched with 15,570 and 43,710 controls, respectively. An additional two cases that were admitted during the washout period between the two waves were included in the overall study group, but not in variant-specific analyses. We were unable to estimate the effectiveness of hybrid and full vaccination immunity against Delta-associated hospitalization, as no infants born to mothers with these immunity statuses were hospitalized during that timeframe. This finding highlights the considerable protective impact of hybrid and full vaccination during the Delta surge. Our analysis indicated that natural immunity effectiveness reached 75% (95% CI: 45–88) and partial vaccination effectiveness was 49% (95% CI: 29–63) during the Delta period. During the Omicron surge, maternal hybrid, natural, and full vaccination immunity exhibited effectiveness rates of 81% (95% CI: 70–88), 48% (95% CI: 25–64), and 64% (95% CI: 52–73), respectively, as compared to the naïve reference group, while partial vaccination did not confer significant protection during this period (Fig. 2).

**Table 1 | Characteristics of the study cohort and infants' hospital admissions with COVID-19 within 180 days of delivery, stratified by cases and controls and by maternal immunization group at the time of delivery (N = 60,121)**

| | Study period [24 August 2021–15 March 2022] | | | Maternal immunization group | | | | | |
| --- | --- | --- | --- | --- | --- | --- | --- | --- | --- |
| | Cases | Controls | *p*-value | Naïve | Natural immunity | Hybrid immunity | Partial vaccination (1–2 doses) | Full vaccination (3–4 doses) | *p*-value- between Immunity groups |
| *N %* | 661 (1.1) | 59,460 (98.9) | | 14,927 (24.8) | 5476 (9.1) | 5852 (9.7) | 21,559 (35.9) | 12,307 (20.5) | – |
| **Maternal age in years** | | | | | | | | | |
| ≤26 | 202 (30.7) | 17,024 (28.7) | 0.260 | 5880 (39.5) | 1963 (35.9) | 1660 (28.5) | 5749 (26.7) | 1974 (16.1) | <0.001 |
| 27–35 | 352 (53.4) | 31,466 (53.1) | | 6728 (45.2) | 2689 (49.2) | 3062 (52.5) | 11,971 (55.7) | 7368 (60.2) | |
| 36–45 | 105 (15.9) | 10,784 (18.2) | | 2280 (15.3) | 814 (14.9) | 1112 (19.1) | 3780 (17.6) | 2903 (23.7) | |
| Multifetal delivery | 22 (3.3) | 2189 (3.7) | 0.744 | 525 (3.5) | 188 (3.4) | 202 (3.5) | 788 (3.7) | 508 (4.1) | 0.042 |
| **Parity** | | | | | | | | | |
| Primipara | 224 (33.9) | 16,823 (28.3) | 0.001 | 4164 (27.9) | 1397 (25.5) | 1395 (23.8) | 6469 (30.0) | 3622 (29.4) | <0.001 |
| Multipara (2–4) | 360 (54.5) | 33,521 (56.4) | | 7759 (52.0) | 2881 (52.6) | 3233 (55.2) | 12,586 (58.4) | 7422 (60.3) | |
| Grandmultipara (5+) | 77 (11.6) | 9116 (15.3) | | 3004 (20.1) | 1198 (21.9) | 1224 (20.9) | 2504 (11.6) | 1263 (10.2) | |
| **Number of SARS-CoV-2 PCR/antigen tests** | | | | | | | | | |
| 0–1 | 157 (23.8) | 18,036 (30.3) | <0.001 | 8023 (53.7) | 859 (15.7) | 641 (11.0) | 6380 (29.6) | 2290 (18.6) | <0.001 |
| 2–4 | 200 (30.3) | 19,486 (32.8) | | 3871 (25.9) | 2400 (43.8) | 2401 (41.0) | 7034 (32.6) | 3980 (32.3) | |
| ≥5 | 304 (46.0) | 21,938 (36.9) | | 3033 (20.3) | 2217 (40.5) | 2810 (48.0) | 8145 (37.8) | 6037 (49.1) | |
| **Newborn characteristics** | | | | | | | | | |
| Female | 317 (48.0) | 29,091 (48.9) | 0.639 | 7286 (48.8) | 2693 (49.2) | 2827 (48.3) | 10,579 (49.1) | 6023 (48.9) | 0.863 |
| Preterm (≤37 weeks GA at delivery) | 45 (6.8) | 3891 (6.5) | 0.760 | 1035 (6.9) | 358 (6.5) | 324 (5.5) | 1348 (6.3) | 871 (7.1) | <0.001 |
| Birthweight ≤2500 gram | 63 (9.5) | 5562 (9.4) | 0.848 | 1482 (9.9) | 503 (9.2) | 461 (7.9) | 1980 (9.2) | 1199 (9.7) | <0.001 |
| **COVID-19 related infant hospitalizations:** | | | | | | | | | |
| *N (% of study group)* | 661 (100) | – | – | 228 (1.5) | 46 (0.8) | 20 (0.3) | 272 (1.3) | 95 (0.8) | |
| Length of stay (days) | 2 (1–3) | – | – | 2 (1–3) | 2 (1–3) | 2 (1–3) | 2 (1–3) | 2 (1–2) | 0.008 |
| Age at COVID-19 admission (days) | 67.5 (±50.6) | – | – | 66.3 (±48.4) | 77.4 (±45.7) | 83.8 (77.9) | 69.6 (±52.8) | 55.7 (±43.1) | 0.057 |
| PICU admission | 33 (5.0) | – | – | 12 (5.3) | 3 (6.5) | 0 (0) | 13 (4.8) | 5 (5.3) | 0.852 |

Data are *n* (%), mean (±standard deviation), median (interquartile range); data are calculated according to the vaccine status of women at delivery. *P*-value was calculated by comparing cases to controls, and between maternal immunity groups using $\chi^2$ test, ANOVA test, Kruskal–Wallis test, as appropriate. All analyses were two-sided.
*GA* gestational age.

## Hybrid and natural immunity effectiveness

Our findings show that maternal hybrid immunity improves infant protection when compared to natural. To further elucidate the potential advantages of hybrid-mediated immunity over natural immunity, we sought to assess the impact of maternal infection timing on infant protection (i.e., first infection was before or during pregnancy; Fig. 3, whole study group, and Supplementary fig. 4A, B for Delta and Omicron periods). The effectiveness trend of hybrid immunity was higher than natural immunity for all measured time points.

Our analysis revealed that the protective effects of hybrid immunity remained comparable for different sequences of hybrid exposure, as shown in Supplementary fig. 5, infection before vaccination (80%, 95% CI: 65–89%) or infection after vaccination (88%, 95% CI: 75–95%). Analysis of hybrid effectiveness based on timing of the last immune stimulation (vaccination or infection), revealed modest

differences in protection. Those born to mothers whose last stimulation occurred in the first 20 weeks of gestation had a slightly lower protection rate of 78% (95% CI 51–90%), while those who received their last stimulation after 20 weeks of gestation had a protection rate of 86% (95% CI 74–92%). These rates resemble those found among the small subgroup of infants (*n* = 342, 2 hospitalized) that were born to mothers whose last stimulation took place before pregnancy 74% (95% CI [−6]–94%) and those whose last stimulation took place during pregnancy (*n* = 5510, 18 hospitalized) 84% (95% CI 74–90%, respectively) (Supplementary fig. 5).

For infants born to mothers with a prior SARS-CoV2 infection, we noted that immune-mediated protection correlated with the number of administered booster doses. Specifically, natural immunity (achieved without vaccination) was found to be effective at a rate of 56% (95% CI 39–68%); hybrid immunity, which ensues following a single vaccine dose, was estimated to be 78% (95% CI 62–87%); and

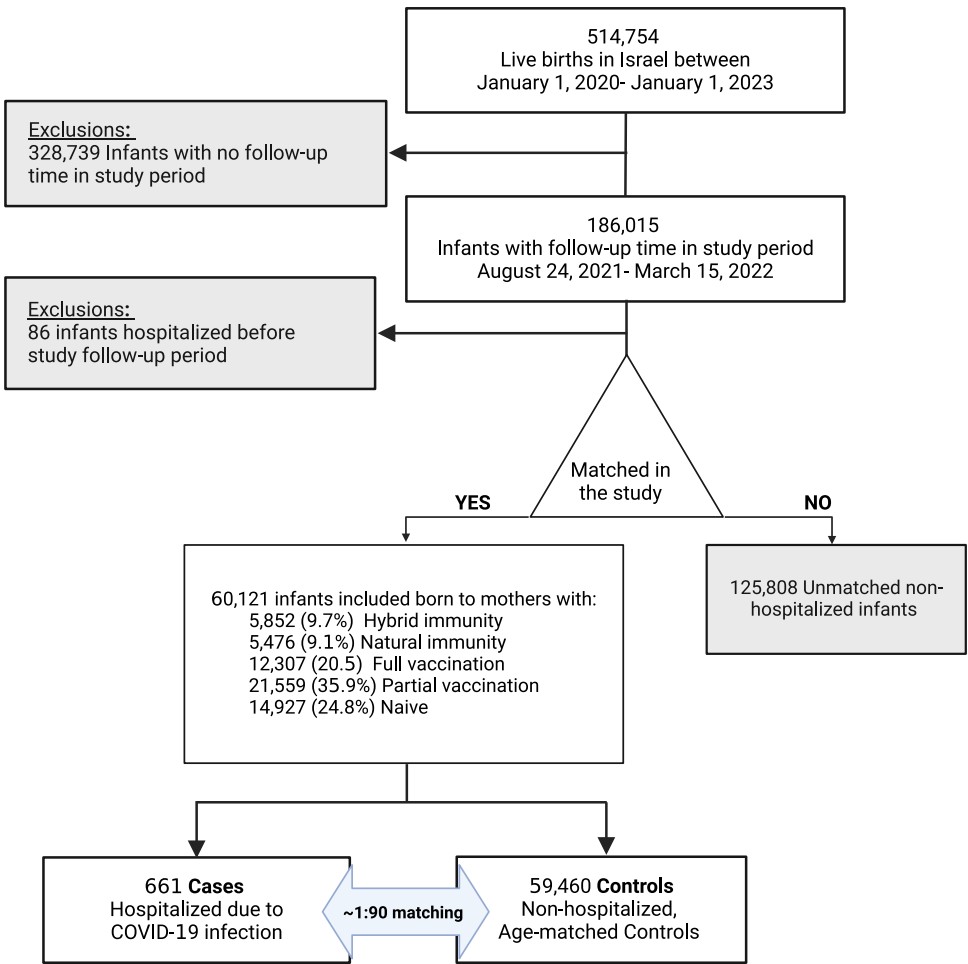

**Fig. 1 | Flow chart of study sample selection.** The flow chart shows the case-control comparison groups investigated during the study period. Created with BioRender.com.

hybrid immunity bolstered by two or more vaccine doses exhibited effectiveness rate of 92% (95% CI 79–97%; Fig. 4).

## Discussion

In this study, we found that maternal hybrid immunity provides heightened protection against infant COVID-19 hospitalization during the first six months of life, as compared to natural immunity. Our analyses show that hybrid immunity effectiveness reached 84% (95% CI, 75–90%) throughout the study period, whereas natural immunity effectiveness was 56% (95% CI, 39–68%). These results emphasize the significance of immunizing women who have recovered from COVID-19 infection during pregnancy to confer robust protection against early infant COVID-19 illness. Infant hospitalizations for COVID-19 were in general brief in all maternal immunity subgroups, and admission to the pediatric ICU was rare. We studied the effect of maternal hybrid immunity on infant COVID-19 hospitalizations, rather than infant SARS-CoV2 infections. Recorded infection rates may be biased by varying testing frequencies among different population groups, particularly among unvaccinated individuals. In Israel, while not consistently implemented across all healthcare facilities, routine SARS-CoV-2 testing of infants presenting with upper respiratory infection symptoms was widely established as a standard procedure in most hospitals during the study period. Given the unbiased approach to testing, and the fact that COVID-19 hospitalization better reflects significant disease burden, we analyzed the data accordingly.

This study concentrated on the indirect benefits of maternal hybrid immunity for unvaccinated newborns. Our findings augment

the growing body of evidence underscoring the considerable protective effects of maternal vaccination during pregnancy on early infant COVID-19 disease. Prior research has primarily focused on vaccinating pregnant women who have not been previously exposed to the virus. A Norwegian study published in 2022 linked maternal vaccination with a reduced risk of infant infection during the first four months of life[18], while a U.S. study associated pregnancy vaccination with a decreased risk of infant hospitalizations and critical disease up to six months of age[12]. Our 2023 study, summarizing data from Israel, demonstrated the significant impact of maternal third booster dose in limiting infant COVID-19 hospitalization[11]. However, clinical data on the durability and benefits of maternal hybrid immunization remain limited. Information on this type of protection stems from studies investigating maternal and newborn protective antibody levels, which indicate high levels of SARS-CoV-2-specific immunoglobulins in maternal blood, cord blood, and milk after vaccinating recovered pregnant women, compared to recovered non-vaccinated pregnant women[13,19,20]. Our study further expands current knowledge with substantial clinical data, illustrating how maternal hybrid immunity translates to early infant protection. Importantly, we demonstrate that hybrid immunization offers greater protection than all other maternal immunization status groups.

Despite significant differences in infant hospitalization rates based on maternal immunity, we observed similar hospitalization durations across all groups, regardless of maternal immunization status. We postulate that the uniformity in stay length may be due to a

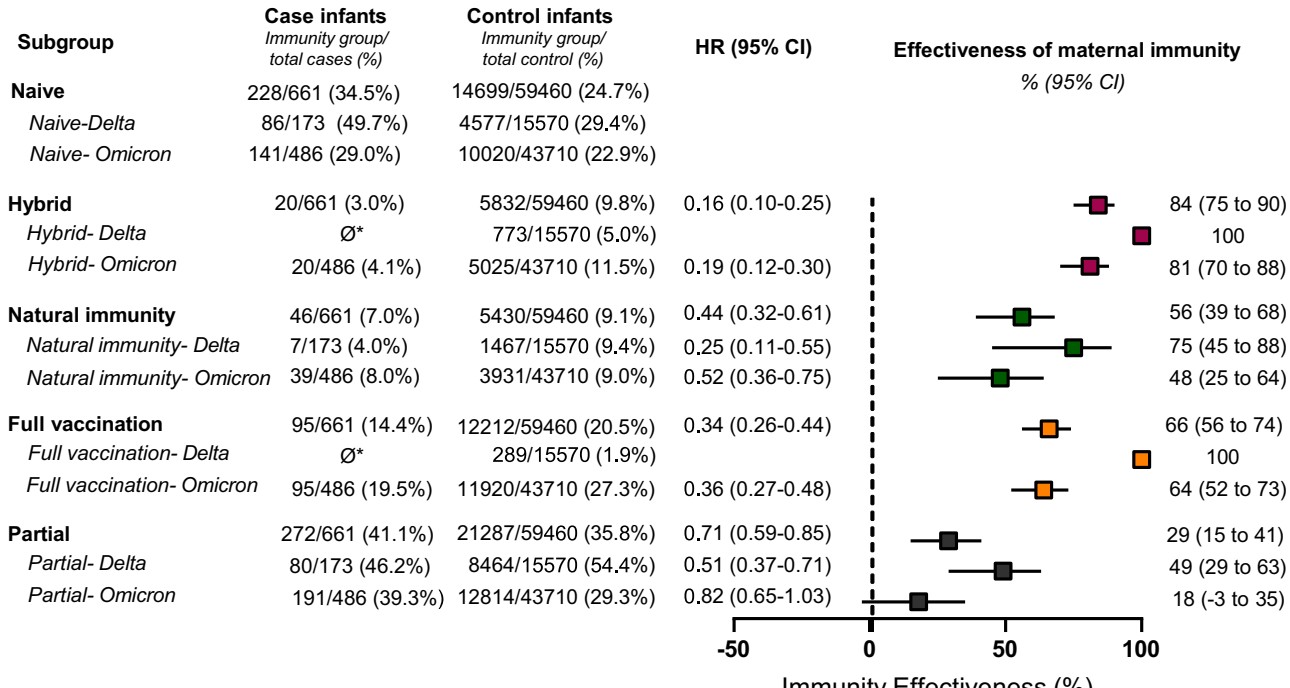

| Subgroup | Case infants _Immunity group/ total cases (%)_ | Control infants _Immunity group/ total control (%)_ | HR (95% CI) | Effectiveness of maternal immunity _% (95% CI)_ |
|---|---|---|---|---|
| **Naive** | 228/661 (34.5%) | 14699/59460 (24.7%) | | |
| _Naive-Delta_ | 86/173 (49.7%) | 4577/15570 (29.4%) | | |
| _Naive- Omicron_ | 141/486 (29.0%) | 10020/43710 (22.9%) | | |
| **Hybrid** | 20/661 (3.0%) | 5832/59460 (9.8%) | 0.16 (0.10-0.25) | 84 (75 to 90) |
| _Hybrid- Delta_ | Ø* | 773/15570 (5.0%) | | 100 |
| _Hybrid- Omicron_ | 20/486 (4.1%) | 5025/43710 (11.5%) | 0.19 (0.12-0.30) | 81 (70 to 88) |
| **Natural immunity** | 46/661 (7.0%) | 5430/59460 (9.1%) | 0.44 (0.32-0.61) | 56 (39 to 68) |
| _Natural immunity- Delta_ | 7/173 (4.0%) | 1467/15570 (9.4%) | 0.25 (0.11-0.55) | 75 (45 to 88) |
| _Natural immunity- Omicron_ | 39/486 (8.0%) | 3931/43710 (9.0%) | 0.52 (0.36-0.75) | 48 (25 to 64) |
| **Full vaccination** | 95/661 (14.4%) | 12212/59460 (20.5%) | 0.34 (0.26-0.44) | 66 (56 to 74) |
| _Full vaccination- Delta_ | Ø* | 289/15570 (1.9%) | | 100 |
| _Full vaccination- Omicron_ | 95/486 (19.5%) | 11920/43710 (27.3%) | 0.36 (0.27-0.48) | 64 (52 to 73) |
| **Partial** | 272/661 (41.1%) | 21287/59460 (35.8%) | 0.71 (0.59-0.85) | 29 (15 to 41) |
| _Partial- Delta_ | 80/173 (46.2%) | 8464/15570 (54.4%) | 0.51 (0.37-0.71) | 49 (29 to 63) |
| _Partial- Omicron_ | 191/486 (39.3%) | 12814/43710 (29.3%) | 0.82 (0.65-1.03) | 18 (-3 to 35) |

**Fig. 2 | Effectiveness of maternal immunization status against infant COVID-19-related hospitalization during the first 180 days of life, stratified according to COVID-19 variants.** Cox proportional hazards models were used to estimate the hazard ratios (HR) and 95% confidence intervals (CI) for COVID-19 hospitalizations in the various groups of maternal immunity status, compared to the maternal naïve group. Models were adjusted for maternal age, gestational age at delivery, parity, and neonatal sex; multifetal gestation, birthweight, and number of maternal documented SARS-CoV-2 tests during pregnancy. Maternal Immunization effectiveness was calculated as (1−adjusted hazard ratio) × 100. Similar analysis was performed when stratified by COVID-19 variants. The delta-predominant period was defined as 24 August 2021 to 1 December 2021. The omicron-predominant period was defined as 15 December 2021 to 15 March 2022. Case infants, _n_ = 661; Control infants, _n_ = 59,460. Boxes and error bars represent the median and 95% CI. Dotted line shows no effect point. Subgroups are indicated by color: Hybrid, red; Natural immunity, green; Full vaccination (3–4 doses), orange; Partial vaccination: 1–2 doses, navy blue.

shared clinical presentation, primarily fever in infants, which necessitates brief hospital stays. However, it is important to note that we lack data regarding the prevalence of fever.

Two studies from 2021 demonstrated that administering the SARS-CoV-2 vaccine during the late second or early third trimester of pregnancy results in increased antibody levels in umbilical cord blood[21,22], and a reduced risk of infant COVID-19 morbidity, compared to earlier gestation vaccination[4,12]. Our previous research supports these findings, revealing that vaccination during the third trimester leads to elevated anti-COVID-19 antibodies in cord blood compared to the first trimester[13]. Our findings align with this observation, indicating that when immune stimulation occurs after 20 weeks of gestation, the level of protection appears to be higher compared to stimulation during the first 20 weeks. However, the difference in protection was somewhat reduced compared to previous reports on vaccination alone[4,12]. It is worth noting that the number of infants born to mothers who were last stimulated before pregnancy was limited in our data, which restricts our ability to accurately predict the true protective effect for infants in that particular group.

Intriguingly, protection resulting from immune stimulation before pregnancy was more robust than that conferred during the first 20 weeks of gestation. This suggests that the durability of immunity acquired through early pregnancy immunization may differ from that obtained pre-pregnancy. Further investigations should focus on these disparities, striving to better understand their origins.

Hybrid-mediated immunity resulted in robust infant protection regardless of the sequence of stimulations (i.e., vaccination before infection or vice-versa). Additionally, our findings suggest that the effectiveness of hybrid-mediated immunity acquired from one vaccine dose is lower than that obtained after two or more doses. Overall these new data provide novel insights into daily clinical questions relevant for patients and families, obstetricians, and healthcare policy. As COVID-19 spread continues, we predict that future guidelines will adopt recommendations for routine SARS-CoV-2 booster vaccination before pregnancy, or during the third trimester[23], aiming to reduce early infant morbidity, similar to recommendations for pertussis and influenza prevention[24–26].

In our analysis, we focus on two key waves of SARS-CoV-2 variants: Delta and Omicron. While Omicron variants have become more prevalent, examining the Delta wave is crucial to understanding the evolving dynamics of the pandemic. This comparison highlights differences between the waves which can be crucial for understanding the importance of updating vaccines to match the currently circulating strains and boosting immunity to overcome waning. However, hybrid immunization enhances protection for both waves.

Our national data reveals a 2.8-fold increase in hospitalization rates among infants aged less than six months during the peak week of Omicron predominance compared to the period of Delta pre-dominance. These results are consistent with CDC reports, which documented a five-fold increase in hospitalization rates when comparing these periods[3]. The observed differences could be due to higher infant infection rates, decreased protection from vaccines and natural immunity, or increased virulence of variants in infants. Our study did not examine the causes underlying these differences: further research is needed to understand these key factors.

Our study has several strengths. We used population-based databases covering the entire Israeli population with high data completeness. In Israel, a substantial proportion of the convalescent pregnant population received the COVID-19 vaccine during pregnancy, providing a large cohort for analysis. The mandatory reporting of

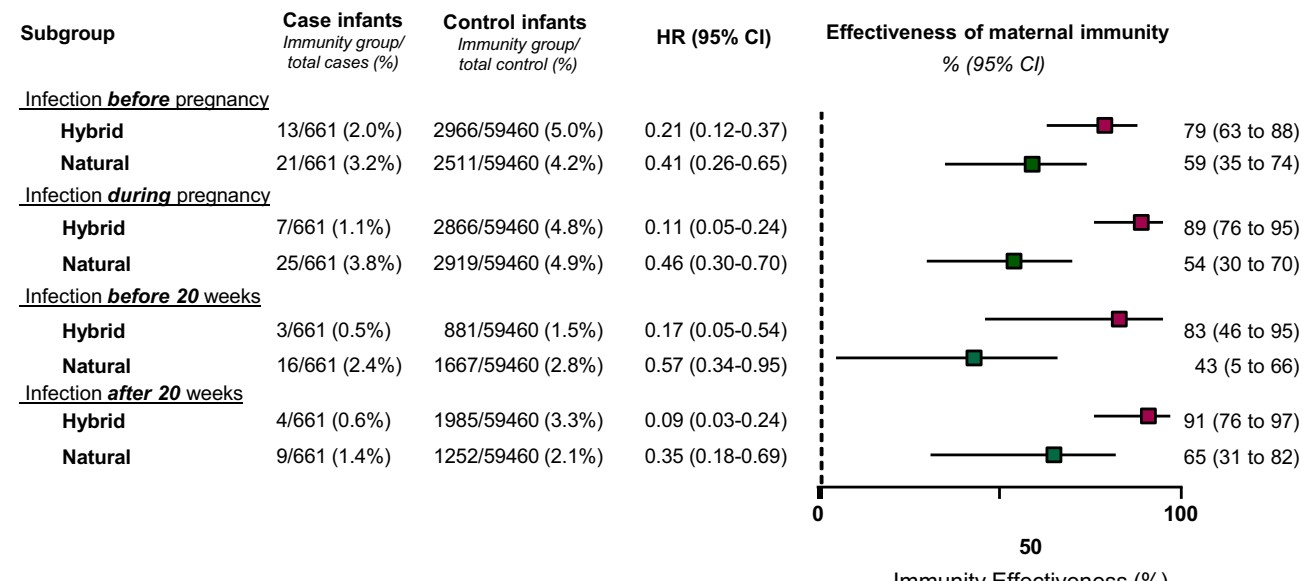

**Fig. 3 | Effectiveness of maternal natural immunity and hybrid immunity against infant COVID-19-related hospitalization during the first 180 days of life, stratified according to maternal COVID-19 infection timing.** Cox proportional hazards models were used to estimate the hazard ratios (HR) and 95% confidence intervals (CI) for COVID-19 hospitalizations in the Hybrid and Natural groups of maternal immunity status, compared to the maternal naïve group, stratified by maternal infection timing. Infection before pregnancy is defined as the last infection that occurred before conception of pregnancy. Infection before 20 weeks refers to the last documented infection that occurred during the first 20 weeks of gestation, while infection after 20 weeks refers to the last documented infection

that occurred during the last 20 weeks of gestation. Models were adjusted for maternal age, gestational age at delivery, parity, and neonatal sex; multifetal gestation, birthweight, and number of maternal documented SARS-CoV-2 tests during pregnancy. Maternal Immunization effectiveness was calculated as (1−adjusted hazard ratio) × 100. Case infants, *n* = 661; Control infants, *n* = 59,460. Boxes and error bars represent the median and 95% CI. Dotted line shows no effect point. Subgroups are indicated by color: Hybrid, red; Natural immunity, green. (Delta and Omicron waves were analyzed separately; results are shown in Supplementary fig. 4A, B).

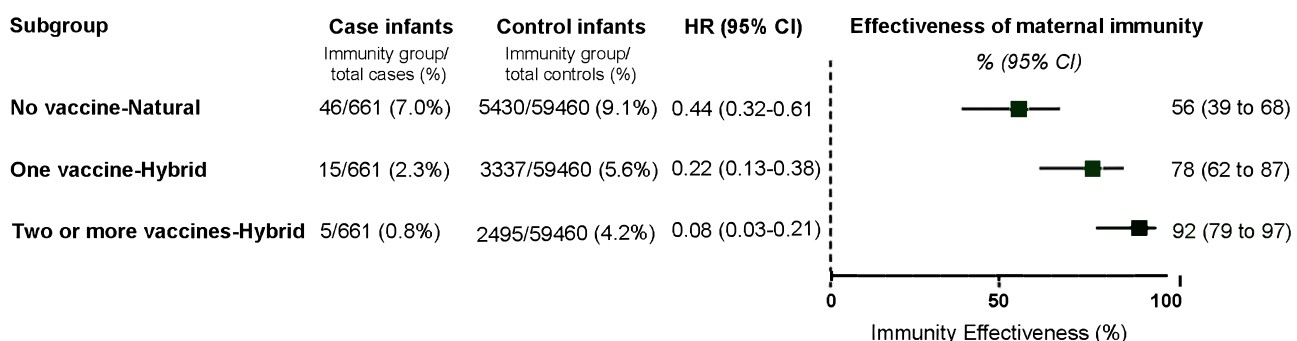

**Fig. 4 | Effectiveness of maternal immunity against infant COVID-19-related hospitalization during the first 180 days of life, stratified according to number of vaccine doses.** Cox proportional hazards models were used to estimate the hazard ratios (HR) and 95% confidence intervals (CI) for COVID-19 hospitalizations according to the number of maternal vaccinations; no vaccination (Natural immunity group); one vaccination (Hybrid group with one vaccination) and ≥2 vaccinations (Hybrid group with two or more vaccination), compared to the

maternal naïve group. Models were adjusted for maternal age, gestational age at delivery, parity, and neonatal sex; multifetal gestation, birthweight, and number of maternal documented SARS-CoV-2 tests during pregnancy. Maternal Immunization effectiveness was calculated as (1−adjusted hazard ratio) × 100. Case infants, *n* = 661; Control infants, *n* = 59,460. Boxes and error bars represent the median and 95% CI. Dotted line shows no effect point.

information to the Israeli national registries, as described in the Methods section, limited the potential for selection bias and provided detailed data on clinical and sociodemographic factors. However, data did not include disease symptoms, constraining any evaluation of severity.

Our investigation presents certain limitations. As a real-world observational study, participants opted for vaccination at different times, making it difficult to account for potential disparities in asymptomatic (and unrecorded) infections, household exposures, health-seeking, behaviors, or risk aversion among individuals.

While we made efforts to adjust for multiple potential confounding factors, the absence of data on critical variables such as

breastfeeding practices, enrollment in daycare centers, and other family or behavioral variables could have influenced the observed outcomes. Additionally, during the study period, changes in testing eligibility occurred, and data on home rapid antigen test results were unavailable. These limitations underscore the complexities of real-world observational research, emphasizing the need for future investigations that incorporate a more comprehensive dataset and account for these variables.

Maternal vaccination during pregnancy offers infant protection through two primary mechanisms: (a) the transfer of protective antibodies via the placenta and breast milk, and (b) a reduction in the infant's exposure to infectious agents due to a better-protected

mother (resulting in decreased vector susceptibility and infectiousness)[27–29]. While it is crucial to comprehensively understand the relative contribution of each protective factor, the necessary data for such determination was not available in this study, highlighting the need for further investigation in future research.

In summary, this real-world assessment revealed that maternal hybrid immunization was associated with a reduced risk of infant hospitalization due to COVID-19 compared to natural immunity via infection alone. Our results show that effectiveness of hybrid immunity remains robust regardless of the timing of stimulation during pregnancy. Our findings supply crucial evidence to reinforce current recommendations advocating for COVID-19 vaccination of previously infected individuals during pregnancy to mitigate substantial illness in early infancy.

## Methods

### Study design, setting, and populations

We conducted a nationwide population-based case-control study in Israel, a country with roughly 9.5 million inhabitants and 190,000 births every year. In Israel, SARS-CoV-2 vaccinations began on 20 December 2020 (1st dose), with the second dose first available from 10 January 2021. At the start of the vaccination drive, groups considered high risk (the elderly, chronic disease patients and health care workers). By this time, pregnant women were considered high risk and were recommended to present for vaccination. The third ("booster") dose rollout began on 30 July 2021. Vaccination of recovered individuals became available to the general population on 24 August 2021. On 15 March 2022, the public policy regarding testing and COVID-19 restrictions was changed. The cases comprised infants who were six months of age or younger during the study period, i.e., who were born between 27 February 2021 and 15 March 2022 and hospitalized due to COVID-19 during the study period. Cases were age matched to non-hospitalized infants (See Flow Chart, Supplementary data- Fig. 1). We excluded infants who were born before 23 weeks of gestation, had a birthweight of <500 grams, missing birthweight, or were hospitalized due to COVID-19 prior to 24 August 2021.

Case infants were defined as those with documented hospitalization due to COVID-19. We collected data on admission and discharge dates and outcomes for all hospitalizations associated with ICD-9 code 07984 (COVID-19 infection) in infants aged 0–180 days. A pediatric expert (Z.E.S.), blinded to preliminary data collection, examined the medical records of infants diagnosed with COVID-19 during their hospital stay, determining whether the primary reason for hospitalization was indeed COVID-19-related. Data on infants who were considered non-eligible as cases (tested positive for COVID-19, but hospitalized for reasons unrelated to COVID-19) are summarized in Supplementary Table 1.

For every case infant, approximately 90 infants were individually matched, based on birthdate (±3 days). Control infants were randomly selected from the birthdate-appropriate population of infants who were not hospitalized for COVID-19 (656 cases had full matching of 90 controls, while 5 cases had 84 matching controls). This approach accounts for both age and timing of exposure to COVID-19 which affect infant morbidity and hospitalization risk. A calendarial distribution of the study cases and controls matching is illustrated in Supplementary fig. 2.

### Immunity status

To evaluate the effectiveness of maternal immunity-conferring events against COVID-19 hospitalization, infants were categorized into five groups based on their mothers' immunity status at the time of delivery: Naïve (i.e., no documentation of SARS-CoV-2 infection or vaccination, Reference Group), natural immunity (i.e., documented positive SARS-CoV-2 test without vaccination), hybrid immunity (i.e., the combination of natural immunity and vaccination with mRNA COVID-19 vaccine), partial vaccination (i.e., vaccination with one or two doses of mRNA COVID-19 vaccine and no infection), and full vaccination (i.e., vaccination with three or four doses of mRNA COVID-19 vaccine and no infection). The mRNA vaccines were BNT162b2-Pfizer and mRNA-1273 Moderna. During the study period, the original monoclonal mRNA vaccine was used; bivalent vaccines were not yet available. Included in the analyses are 5 women who received non-mRNA vaccination.

### Data source and organization

Data were retrieved from TIMNA, a national research infrastructure for big data established by the Israel Ministry of Health (MOH) to facilitate large-scale health research. The MOH maintains a COVID-19 registry, which compiles information on all laboratory-based SARS-CoV-2 diagnostic tests, vaccinations, and confirmed cases. Our study analyzed integrated participant-level data provided by the MOH from three main sources: the National Birth Registry, the COVID-19 Registry, and the National Inpatient Registry, which encompasses all diagnoses of hospitalized patients in Israel.

### Statistical analysis

Descriptive statistics of the study population (cases vs. controls) were presented as proportions or means, as appropriate. Differences between groups were calculated using $\chi^2$, and ANOVA with post hoc Bonferroni analysis as appropriate; length-of-stay was analyzed with Kruskal–Wallis test. In this matched case-control study, data analyses employed stratified Cox regressions, where each case and its controls accounted for one stratum, yielding weighted HR which is an estimate for the odds ratios in conditional logistic regressions. Models were adjusted for maternal age, gestational age at delivery, parity, and neonatal sex; multifetal gestation, birthweight categories (<2500, 2500–3999, ≥4000) and number of maternal documented SARS-CoV-2 tests categories during pregnancy (0–1, 2–4, ≥5). We further analyzed these associations by dominant-variant period, having separate models for the Delta (24 August 2021–1 December 2021) and Omicron (15 December 2021–15 March 2022) periods, and excluding the washout period between the variants (2 December 2021–14 December 2021). Maternal mediated immunity effectiveness was estimated as a percentage, defined as (1−HR) × 100; 95% CI were calculated similarly, for the entire study period and for the Delta and Omicron periods separately.

We conducted additional analyses stratified by timing of immunity conferring events (i.e., infection or vaccination before or during pregnancy). Among infants of the hybrid group, we also evaluated the association between the sequence of immunity-conferring events (infection before or after vaccination), timing of last stimulation (i.e., vaccination or infection) for three different time groups: before pregnancy, during pregnancy up to 20 weeks of gestation, and during pregnancy ≥20 weeks of gestation). For infants born to mothers with previous infections we estimated the association between the number of vaccine doses (i.e., none vs. one vs. two or more) and infant COVID-19 hospitalizations. We compared the effectiveness of each maternal immunity subgroup to the Naïve (reference) group by Cox proportional hazard modeling. In order to evaluate the effectiveness of each type of maternal immunity compared to the others, i.e. Hybrid vs. Natural immunity, the HRs of the maternal immunity groups were compared using p-values derived from z-scores according to the following equation (Equation 1):

$$Z = \frac{\beta(0) - \beta(\times)}{\sqrt{SE(0)^2 + SE(\times)^2}}$$

Where (0) represents the regression coefficient and its standard error for the hybrid immunity, and (x) represents these parameters in any other immunity group used for the comparison.

Python version 3.7.3 and lifelines 0.24.14 were used for Cox models. IBM-SPSS for Windows, version 29 (IBM Corp., Armonk, N.Y., USA), was used for descriptive and univariate analyses. A two-sided *p*-value ≤ 0.05 was considered to indicate statistical significance in all analyses. A *p*-value ≤ 0.05 was considered to indicate statistical significance in all analyses. The study protocol was approved by the Hadassah Medical Organization's Institutional Review Board (Helsinki Committee approval #0593-21, 5 September 2021). The committee granted exemption from informed consent, based on preserving the participants' anonymity.

## Reporting summary

Further information on research design is available in the Nature Portfolio Reporting Summary linked to this article.

## Data availability

The data that support the findings of this study are available from the authors but due to national and organizational data privacy regulations, individual-level data such as those used for this study cannot be shared openly. Restrictions apply to the availability of these data, which were used under license from the Israel Ministry of Health for the current study, and so are not publicly available. Data are, however, available from the ministry upon request and with permission from the Israel Ministry of Health https://govextra.gov.il/ministry-of-health/big-data-research/home/.

## Code availability

The modeling in this paper used Python version 3.7.3 and lifelines 0.24.1, which are freely available.

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

## Acknowledgements
This study was supported by The Magda and Richard Hoffman Center for Human Placental Research (M.L.; S.M.C.; D.W.; and O.B.) and the "Ofek" Program of the Hadassah Medical Center (O.B.). The Jerusalem Center for Personalized Computational Medicine provided post-doctoral support (J.G.).

## Author contributions
J.G., M.L., Y.S., E.M., A.C.P. and O.B. saw the original data, collected it and analyzed it. J.G., M.L., O.B., R.C.M., G.S. and S.Y. conceived and designed the study. Z.E.S. reviewed the neonatal data and outcomes. J.G., M.L., O.B., R.C.M., S.M.C., Y.S., D.G.-W., A.C.P. and S.Y. wrote the manuscript. All authors critically reviewed the manuscript and decided to proceed with publication. R.C.M., S.Y. and O.B. supervised the study process. O.B. vouches for the data and analysis. E.M. and G.S. combined, anonymized, and QC of the MOH data.

## Competing interests
The authors declare no competing interests.

## Additional information

[1]Braun School of Public Health, Hadassah Medical Center, Faculty of Medicine of the Hebrew University of Jerusalem, Jerusalem, Israel. [2]Obstetrics & Gynecology Division, Hadassah Medical Center, Faculty of Medicine of the Hebrew University of Jerusalem, Jerusalem, Israel. [3]Henrietta Szold Hadassah Hebrew University School of Nursing in the Faculty of Medicine Jerusalem, Jerusalem, Israel. [4]The Jerusalem Center for Personalized Computational Medicine Jerusalem, Jerusalem, Israel. [5]TIMNA-Israel Ministry of Health's Big Data Platform, Israel Ministry of Health, Jerusalem, Israel. [6]Neonatology Department Hadassah Medical Center, Faculty of Medicine of the Hebrew University of Jerusalem, Jerusalem, Israel. [7]These authors contributed equally: Joshua Guedalia, Michal Lipschuetz. [8]These authors jointly supervised this work: Ronit Calderon-Margalit, Ofer Beharier. ✉e-mail: Michal.lipschuetz@gmail.com; oferbeharier@gmail.com

