## [Peer Review File · Nature Communications]

Maternal hybrid immunity and risk of infant COVID-19 hospitalizations: national case-control study in IsraelREVIEWER COMMENTS

Reviewer #1 (Remarks to the Author):

- No COVID-19 vaccine is authorized for infants <6 months of age. This report attempts to quantify the effectiveness of various maternal immunities to COVID-19 against infant hospitalization in the first 6 months of life during the delta and Omicron waves of infection in Israel. The findings highlight that hybrid immunity is superior to vaccination or infection alone which is a novel and important finding.

A few specific comments:

- Why is 3 doses considered full vaccination as this would include a booster dose. To better understand this choice, it would be helpful to include the recommended time period between dose #2 and #3 during Israel at the time of the study. There should also be more information about vaccine availability to understand to whom vaccines were available before August 24, 2021 and how the group that had vaccines available before August 24, 2021 may be different in COVID-19 risk than the general population, e.g., healthcare workers? There should also be some comment on how might this bias results.
- Multiple studies have shown decreased protection of vaccination during Omicron versus Delta time periods, however much of this time this difference is related to waning immunity. Some discussion about time since last stimulation (infection or vaccination) may be warranted in this manuscript or that there was lack of information about this.
- Is it possible to make more explicit whether the findings of this study can inform recommendations for when in pregnancy to recommend COVID vaccines.
- Is it possible to look at product specific VE?
- It should be specified that this was the original monoclonal vaccine and bivalent vaccines were not available during the study period.
- What does "infectious" mean in supplemental table 1 as a main medical diagnosis and why were these excluded?
- Line 97, should, "Cases and controls" be deleted?
- Line 100 - unnecessary return/paragraph
- Study time period start is listed as August 24, 2021, however infants born between February 27, 2021 - March 15, 2022 were included. Why not just state that the study included infants <6 months of age between August 24, 2021 - March 15, 2022.

Reviewer #2 (Remarks to the Author):

This study is one of many to address the role of COVID-19 vaccination in pregnant women., I am unclear from the introduction or conclusion what is new about this study compared to the many papers published in this area. I have concerns regarding study design, use of causal language and statistical analyses

Major issues

1. language is important - you cannot assess the effectiveness of hybrid immunity using a case-control study design. Please amend to appropriate language.
2. why conduct a case control study when figure 1 shows that you could have conducted a cohort study?
3. This manuscript requires a review by a statistical expert. In the methods section it states that all findings are compared to the maternal naive group - however in the main body of the paper and the conclusions the comparisons are between hybrid and natural immunity. The authors do not present those comparisons in the results.
4. one of the main findings of this study is that for those infants aged under 6 months of age if they require hospitalisation for covid-19 it is likely to be a short admission. Even admission to ICU was rare in this hospitalised group.
5. there was a larger proportion of primiparous mothers in the cases group - admission may be related to health seeking behaviour rather than driven by clinical need - this is not addressed or discussed in the paper.
6. there is limited discussion of the limitations of the study design

Reviewer #3 (Remarks to the Author):

This is a case control study that leverages a national database to assess whether the timing of immune-conferring events to pregnant mothers has a measurable impact on downstream hospitalizations of their eventual infants. They do. Ultimately, these data support vaccine-stimulation during pregnancy, even for those with so-called natural immunity.

The data are impressive in their completeness and size, owing to the success of Israel's epidemiological advantages (that is, an enviable patient database and a cohort of able scientists able to query and analyze its findings).

The figures and tables are fairly clear, although it would be nice to see the hazard ratios, because the unadjusted numerators and denominators do not remotely match the vaccine effectiveness calculations from the raw numbers. So, it's hard to "check the work" as I often like to do just to make sure everything is as it is "supposed to be."

The manuscript, on the other hand is extremely difficult to follow. I did not understand the Abstract the first time I read it. After reading the entire study, I understand the Abstract. But an Abstract really needs to be written in such a clear way that I know what was done and what was found, even if I read no further. As it stands, the Abstract now confirms what I gleaned after a close read of the paper—so that's good. But I honestly was confused by it as a stand-alone paragraph. We need to know the timing of events. We need very clear definitions up front.

In fact, the manuscript itself continues in this pattern. Most of the sentences make sense in isolation, but at many points, I got lost; "Wait who is this?" I kept saying and "wait when did this happen?" etc. I spent a lot of time being confused and having to flip back and forth to the figures/tables to clear up what was even happening.

The big problem is definitions and timing. A much clearer early description of the controls is needed. The primary problem, I believe, is that inadequate definitions are given as we go. The Abstract says the controls were not hospitalized. But they had to be, otherwise what are we measuring? (They were not hospitalized for COVID in particular, as the flow chart shows).

Any time an immune-conferring event is mentioned, we need to know whether that was pre-pregnancy or during (and if during, early or late when applicable). Otherwise, I just kept on trying to figure out "was this during or before?" when looking at vaccination or natural infection. Again, I was just constantly teetering between being confused and then having things cleared up when I flipped back to some other page and figured it out.

A point about the data: the VE calculation in a study like this is not that straightforward; what are we measuring, and where is the numerator and denominator for each calculation derived. It would be best if we were given clear examples using numbers from the study in the Methods. Overall, it's just really hard to see where the numbers come from, especially since adjusted HRs are never given.

Lastly, I am unsure why we need any Delta data at this point. Omicron and its variants are all that is out there right now.

Small points:

Line 222: what is natural immunization?

Line 231: I'm having a lot of trouble understanding how this is possible, especially given the ample data on the waning of immunity.

Supplemental Figure 2: It would be very helpful to have a share graph of this. That is, two more panels that have the share (adding to 100% for each the cases and the controls) for each of the types of maternal immunity

Supplemental Figure 3: the wide confidence interval on the "last stimulation before pregnancy" is notable. In general the LOW END of the confidence intervals seems to support the notion that

“more recent/later in pregnancy vaccine-conferred immunity” is better. The point estimates are not that different. But the lower thresholds seem to tell a story here that is worth attention in each of the “pairs”

Reviewer #4 (Remarks to the Author):

It is a very interesting manuscript about the effectiveness of the offspring of previous exposure to COVID-19 in the mother (either vaccination or infection). Showing moderate-to-high levels of protection. I have some concerns about the methods used for the analysis.

My main concern with the paper is the lack of information about the number of doses for the mother after the delivery. Suppose the naïve mother is vaccinated after delivery. In that case, it can bias the estimates downwards, like the other groups, but biasing upwards, considering the presence of IgA and IgG in human milk (<https://bmcpregnancychildbirth.biomedcentral.com/articles/10.1186/s12884-022-04945-z> | [https://www.thelancet.com/journals/eclinm/article/PIIS2589-5370\(22\)00123-7/fulltext#seccesectitle0001](https://www.thelancet.com/journals/eclinm/article/PIIS2589-5370(22)00123-7/fulltext#seccesectitle0001)). The authors should describe the mothers' vaccination status after the delivery, if the unvaccinated became vaccinated, etc.

Other points:

It is necessary for a better description of the survival analysis. How was the matching procedure accounted for in the Cox regression? As it is a case-control study, the matching strata can't be ignored.

Did the authors check for proportional hazards? In the presence of covariates and non-proportional hazards, the standard error is misestimated and needs to be estimated using Bootstrap. (<https://doi.org/10.1001/jama.2020.1267>)

In the discussion Line 247. There is information about a 2.8-fold increase in hospitalisation without reference to a text or paper.

The length of stay is similar across all groups. Do the authors have any hypotheses about it? A paragraph about it should enrich the discussion.

Line 118-127. Length of stay tends to be very right-skewed, so a median (interquartile range) should be described better.

Line 153 (84%, 95% CI: 75-90% vs. 56%, 95% CI: 39-68%, $p < 0.001$) – Is this p-value from a contrast between maternal hybrid versus natural groups?

A histogram showing the age, in days, of the hospitalisation of the cases is helpful. As the study period covers only 206 days, it should have a very low number of individuals, between 150-180 days.

Please change phrases using “recent” to the publication year so we can grasp the temporal association. (In discussion)

The discussion lacks a paragraph about the limitations of the study.

What test was applied to the Bonferroni correction? (Methods line 335), if not used, remove it from methods.

Methods: Line 291 states the study start is February 27, 2021; however, all other parts of the text state August 24.

Supplementary Figure 2: It lacks the scale of the axis Y

Code availability: Please provide the code used in the analysis in an open repository (zenodo, GitHub, etc..)

REVIEWER COMMENTS.

Reviewer #1 (Remarks to the Author):

- No COVID-19 vaccine is authorized for infants <6 months of age. This report attempts to quantify the effectiveness of various maternal immunities to COVID-19 against infant hospitalization in the first 6 months of life during the delta and Omicron waves of infection in Israel. The findings highlight that hybrid immunity is superior to vaccination or infection alone which is a novel and important finding.

A few specific comments:

- Why is 3 doses considered full vaccination as this would include a booster dose. To better understand this choice, it would be helpful to include the recommended time period between dose #2 and #3 during Israel at the time of the study. There should also be more information about vaccine availability to understand to whom vaccines were available before August 24, 2021 and how the group that had vaccines available before August 24, 2021 may be different in COVID-19 risk than the general population, e.g., healthcare workers? There should also be some comment on how might this bias results.

Answer: In Israel during the study period, pregnant women were considered a high-risk group and invited to present for vaccination as soon as vaccines became available. Specifically, SARS-CoV-2 vaccinations began on 20 December, 2020 (1st dose), with the second dose first available from 10 January 2021. At the start of the vaccination drive, groups considered high risk (the elderly, chronic disease patients and health care workers). By this time, pregnant women were considered high risk and were recommended to present for vaccination. The third (“booster”) dose rollout began on 30 July 2021. **Change appears in Methods, p. 14.**

- Multiple studies have shown decreased protection of vaccination during Omicron versus Delta time periods, however much of this time this difference is related to waning immunity. Some discussion about time since last stimulation (infection or vaccination) may be warranted in this manuscript or that there was lack of information about this.

Thank you for raising these interesting points. Referring to Figure 2, we showed the differences between the Delta and Omicron waves in all exposed subgroups, as compared to the Naïve reference group. To determine the relative effectiveness of stimulation occurring before or during pregnancy, and before or after 20 weeks’ gestation (i.e. closer or more remote from delivery) during the Delta and Omicron periods, we further analyzed these data. Please see Figure 3 and Suppl Figure 4A-B As the figures show, regardless of the timing of last stimulation prior to or during pregnancy, protection was greater during the Delta wave. Indeed, there were no cases of infant hospitalizations among mothers in the Hybrid immunity group, during the Delta wave.

We have added reference to the extended figure to the Results section (p. 8) and the Discussion section. **Changes appear on p. 7 and p.11 and suppl figure 4A-B.**

- Is it possible to make more explicit whether the findings of this study can inform recommendations for when in pregnancy to recommend COVID vaccines.

Thank you for this important comment: unlike vaccination in naïve populations, which was shown to be more effective when delivered in the third trimester, our results showed that hybrid immunity provides similar infant protection regardless of timing or sequence during pregnancy. **Change appears in the summary on p. 13.**

Is it possible to look at product specific VE?

Among our population, only 5 women received non-mRNA vaccination. They were included in analyses. **Change appears on p. 15.**

It should be specified that this was the original monoclonal vaccine and bivalent vaccines were not available during the study period.

During the study period, the original monoclonal RNA vaccine was used; bivalent vaccines were not yet available.

These two points were added to the Participants subsection, **change appears on p. 15.**

- What does "infectious" mean in supplemental table 1 as a main medical diagnosis and why were these excluded?

Answer: We have revised the table and added a list of the infectious diseases (non-COVID) suffered by these infants, which were the main reason for their hospitalizations, and the reason for their exclusion from the study group. **Change appears in the Table.**

- Line 97, should, "Cases and controls" be deleted.

Done.

- Line 100 - unnecessary return/paragraph.

Done.

- Study time period start is listed as August 24, 2021, however infants born between February 27, 2021 - March 15, 2022 were included. Why not just state that the study included infants <6 months of age between August 24, 2021 - March 15, 2022.

While our cases (i.e., hospitalized infants) were identified according to their dates of birth and hospitalization, the study considered the immune status of their mothers during pregnancy. Therefore, we described the study period from the earliest time the mother-

infant dyad could be included, from the time that vaccines became available. Change appears on page 14.

Reviewer #2 (Remarks to the Author):

This study is one of many to address the role of COVID-19 vaccination in pregnant women., I am unclear from the introduction or conclusion what is new about this study compared to the many papers published in this area. I have concerns regarding study design, use of causal language and statistical analyses

Major issues

1. language is important - you cannot assess the effectiveness of hybrid immunity using a case-control study design. Please amend to appropriate language.

Answer: We appreciate the reviewer's query and the chance to elaborate on these issues. Firstly, the novelty of our study lies in our focus on hybrid immunity, whereas most earlier large studies addressed the impact of vaccination alone. The clinical question of if and when to vaccinate a recovered pregnant patient, remains open, at a stage when large proportion of pregnant patients have past infection with COVID-19.

As regards the appropriate language, we have made changes throughout to highlight that are results show estimated effectiveness.

2. why conduct a case control study when figure 1 shows that you could have conducted a cohort study?

We contemplated various study designs. Our study is complicated as we have two types of exposures - vaccination and COVID-19 illness. Each of these exposures could happen at a different time point – both calendrical and in relation to the pregnancy. To complicate things even further – they could vary in terms of which exposure precedes the other. A "simple" time dependent exposure would not work in this case (we have two time-dependent exposures) and would not take into account the background morbidity waves of COVID-19. We considered multiple approaches and found biases in all of them. To overcome these biases, we decided to adopt a nested case control design, matching cases to controls based on date of birth, thus controlling for external exposure during pregnancy. We find this design to be the simplest and easiest to understand, with the fewest potential biases of all designs. In these analyses, time of follow up is ignored.

It should be noted that a similar design has been implemented in other studies of the association between maternal vaccination and offspring disease. There are ample studies (a quick search shows about 3000 entries in Pubmed) estimating vaccine effectiveness using a case-control design, including studies of the validity of the design that support our decision to adopt this design.

Shapiro, Eugene. Case-control studies of the effectiveness of vaccines: validity and assessment of potential bias. *Pediatr Infect Dis J.* 2004;**23(2)**:127-131. Cited in: Journals@Ovid Full Text at

<http://ovidsp.ovid.com/ovidweb.cgi?T=JS&PAGE=reference&D=ovftg&NEWS=N&AN=00006454-200402000-00008>. Accessed October 26, 2023.

Franke MF, Jerome JG, Matias WR, et al. Comparison of two control groups for estimation of oral cholera vaccine effectiveness using a case-control study design. *Vaccine*. 2017 Oct 13;35(43):5819-5827. doi: 10.1016/j.vaccine.2017.09.025. Epub 2017 Sep 12. PMID: 28916247; PMCID: PMC5661944.

And also including studies of COVID-19 vaccine:

Simeone RM, Zambrano LD, Halasa NB, et al., Overcoming COVID-19 Investigators. Effectiveness of Maternal mRNA COVID-19 Vaccination During Pregnancy Against COVID-19-Associated Hospitalizations in Infants Aged <6 Months During SARS-CoV-2 Omicron Predominance - 20 States, March 9, 2022-May 31, 2023. *MMWR Morb Mortal Wkly Rep*. 2023 Sep 29;72(39):1057-1064. doi: 10.15585/mmwr.mm7239a3. PMID: 37874864.

Halasa NB, Olson SM, Staat MA, et al., Overcoming Covid-19 Investigators. Maternal Vaccination and Risk of Hospitalization for Covid-19 among Infants. *N Engl J Med*. 2022 Jul 14;387(2):109-119. doi: 10.1056/NEJMoa2204399. Epub 2022 Jun 22. PMID: 35731908; PMCID: PMC9342588.

The matched cohort study from Scotland also employed a similar approach, ignoring time of follow up:

Lindsay L, Calvert C, Shi T, et al., Neonatal and maternal outcomes following SARS-CoV-2 infection and COVID-19 vaccination: a population-based matched cohort study. *Nat Commun*. 2023 Aug 29;14(1):5275. doi: 10.1038/s41467-023-40965-9. PMID: 37644002; PMCID: PMC10465539.

3. This manuscript requires a review by a statistical expert. In the methods section it states that all findings are compared to the maternal naive group - however in the main body of the paper and the conclusions the comparisons are between hybrid and natural immunity. The authors do not present those comparisons in the results.

We have added text to clarify our methods. We compared the effectiveness of each maternal immunity subgroup to the Naïve (reference) group by Cox proportional hazard modeling. In order to evaluate the effectiveness of each type of maternal immunity compared to the others, i.e. Hybrid vs. Natural immunity, the HRs of the maternal immunity groups were compared using p-values derived from z-scores.

We have now added the above to the Methods section (page 17) and have edited the Results section to clarify which methodological approach was used to obtain the findings (changes throughout).

4. one of the main findings of this study is that for those infants aged under 6 months of age

if they require hospitalisation for covid-19 it is likely to be a short admission. Even admission to ICU was rare in this hospitalised group.

This is true, and an important point. We have added a line to the Discussion (pages 9, 10): Infant hospitalization for COVID-19 was in general brief in all maternal immunity groups, and admission to the PICU was rare.

5. there was a larger proportion of primiparous mothers in the cases group - admission may be related to health seeking behaviour rather than driven by clinical need - this is not addressed or discussed in the paper.

This is an interesting point; the Cox regression that we reported, controlled for parity to take this into consideration. To further elucidate the potential effect of parity on infant hospitalization, an additional analysis was performed comparing the rate of primiparous vs. parous women among cases and controls. This showed that there was no apparent difference between the hazard ratio and effectiveness between primiparas and multiparas among Hybrid vs. Naïve or Natural immunity vs. Naïve groups, as seen in the Table, below, which were the groups that formed the main focus of our work.

	Primipara		Multipara	
	HR	immunity effectiveness	HR	immunity effectiveness
Hybrid	0.17 (0.07-0.38)	83% (62-93%)	0.17 (0.09-0.29)	83% (71-91%)
Natural immunity	0.44 (0.26-0.77)	56% (23-74%)	0.42 (0.28-0.63)	58% (37-72%)
Full vaccination	0.26 (0.16-0.42)	74% (58-84%)	0.39 (0.28-0.53)	61% (47-72%)
Partial	0.53 (0.39-0.73)	47% (27-61%)	0.80 (0.63-1.01)	20% ([-1]-37%)

6. there is limited discussion of the limitations of the study design

We have added descriptions of limitations to the study to the Discussion section. Changes appear on pp. 12-13.

Reviewer #3 (Remarks to the Author):

This is a case control study that leverages a national database to assess whether the timing of immune-conferring events to pregnant mothers has a measurable impact on downstream hospitalizations of their eventual infants. They do. Ultimately, these data support vaccine-stimulation during pregnancy, even for those with so-called natural immunity.

The data are impressive in their completeness and size, owing to the success of Israel's epidemiological advantages (that is, an enviable patient database and a cohort of able scientists able to query and analyze its findings).

We thank the reviewer for their careful reading of our manuscript and their supportive remarks.

The figures and tables are fairly clear, although it would be nice to see the hazard ratios, because the unadjusted numerators and denominators do not remotely match the vaccine effectiveness calculations from the raw numbers. So, it's hard to "check the work" as I often like to do just to make sure everything is as it is "supposed to be."

We thank the reviewer for this comment. We were not sure we understood what changes to the tables and figures were required. The Hazard ratios (HR) were calculated and added to figures 2-4 and suppl figures 4A-B and 5.

Unadjusted HRs – calculated without inclusion of potential confounders – appear below:

	Unadjusted HR	Unadjusted HR lower 95 CI	Unadjusted HR upper 95 CI
Hybrid	0.2	0.13	0.32
Natural immunity	0.54	0.39	0.74
Full vaccination	0.45	0.35	0.57
Partial	0.85	0.71	1.01

The manuscript, on the other hand is extremely difficult to follow. I did not understand the Abstract the first time I read it. After reading the entire study, I understand the Abstract. But an Abstract really needs to be written in such a clear way that I know what was done and what was found, even if I read no further. As it stands, the Abstract now confirms what I gleaned after a close read of the paper—so that's good. But I honestly was confused by it as a stand-alone paragraph. We need to know the timing of events. We need very clear definitions up front.

In fact, the manuscript itself continues in this pattern. Most of the sentences make sense in isolation, but at many points, I got lost; "Wait who is this?" I kept saying and "wait when did this happen?" etc. I spent a lot of time being confused and having to flip back and forth to the figures/tables to clear up what was even happening.

The abstract was thoroughly revised with an eye to clarifying what was done and the results. We also reviewed the Methods and Results sections to further clarify our approach and resulting findings. Changes appear throughout the Abstract and Methods.

The big problem is definitions and timing. A much clearer early description of the controls is needed. The primary problem, I believe, is that inadequate definitions are given as we go. The Abstract says the controls were not hospitalized. But they had to be, otherwise what are we measuring? (They were not hospitalized for COVID in particular, as the flow chart shows).

As mentioned in our response to the previous comment, we have endeavored to clarify all these points. Controls were infants that were not hospitalized for COVID-19. They were matched to the cases by birthdate ± 3 days, at $\sim 1:90$ ratio. Changes appear in Figure 1 and the abstract.

Any time an immune-conferring event is mentioned, we need to know whether that was pre-pregnancy or during (and if during, early or late when applicable). Otherwise, I just kept on trying to figure out “was this during or before?” when looking at vaccination or natural infection. Again, I was just constantly teetering between being confused and then having things cleared up when I flipped back to some other page and figured it out.

As mentioned, above, we have endeavored to clarify the Methods and Results sections. Changes appear throughout.

A point about the data: the VE calculation in a study like this is not that straightforward; what are we measuring, and where is the numerator and denominator for each calculation derived. It would be best if we were given clear examples using numbers from the study in the Methods. Overall, it’s just really hard to see where the numbers come from, especially since adjusted HRs are never given.

We have added the adjusted Hazard Ratios to the Figures. Column headings were also edited to clarify the numerators and denominators in each analysis. Changes appear in Figures 2-4 and extended figures 4A-B and 5.

Lastly, I am unsure why we need any Delta data at this point. Omicron and its variants are all that is out there right now.

True, Omicron and its variants are the dominant strains now, however, we believe that analyzing data from the Delta period as well, augments our understanding regarding dynamics of the pandemic and alterations over time of vaccine effectiveness as well as waning. Therefore, we have added the following to the Discussion to clarify the importance of investigating and comparing with the Delta strain:

In our analysis, we focus on two key waves of SARS-CoV-2 variants: Delta and Omicron. While Omicron variants have become more prevalent recently, examining the Delta wave is crucial to understanding the evolving dynamics of the pandemic. This comparison highlights differences between the waves, particularly in virulence and vaccine effectiveness. Compared to the Delta variant, Omicron was less virulent but spread faster. Moreover, although mRNA vaccines, initially formulated for the wild type variant, maintained their effectiveness against Delta, their efficacy seems to have decreased against Omicron, possibly due to its distinct mutations and greater waning effect. These observations are crucial for understanding the importance of updating vaccines to match the currently circulating strains and boosting immunity to overcome waning.

Change appears on p. 11.

Small points:

Line 222: what is natural immunization?

Thank you for drawing our attention to this misprint: sentence is now corrected, p. 10.

Line 231: I'm having a lot of trouble understanding how this is possible, especially given the ample data on the waning of immunity.

Thank you for raising this important point. We speculate that this observation may stem from modulation of the immune response in early pregnancy. We have added a sentence to that effect, supported by relevant references. Change appears on p. 11 and references 23-24.

Supplemental Figure 2: It would be very helpful to have a share graph of this. That is, two more panels that have the share (adding to 100% for each the cases and the controls) for each of the types of maternal immunity

We have added 3 charts to Supplemental Figure 2 (C-E), comparing the proportions of maternal status at birth among cases and controls up to 100% over the study period, and a bar chart showing the aggregate proportions of each subgroup. Change appears in former Suppl Figure 2, now Suppl Figure 3 (3C-E).

Supplemental Figure 3: the wide confidence interval on the "last stimulation before pregnancy" is notable. In general the LOW END of the confidence intervals seems to support the notion that "more recent/later in pregnancy vaccine-conferred immunity" is better. The point estimates are not that different. But the lower thresholds seem to tell a story here that is worth attention in each of the "pairs"

Thank you for this insight. Unfortunately, we do not have sufficient numbers in this study to answer this question.

Reviewer #4 (Remarks to the Author):

It is a very interesting manuscript about the effectiveness of the offspring of previous exposure to COVID-19 in the mother (either vaccination or infection). Showing moderate-to-high levels of protection. I have some concerns about the methods used for the analysis.

My main concern with the paper is the lack of information about the number of doses for the mother after the delivery. Suppose the naïve mother is vaccinated after delivery. In that case, it can bias the estimates downwards, like the other groups, but biasing upwards, considering the presence of IgA and IgG in human milk

(<https://bmcpregnancychildbirth.biomedcentral.com/articles/10.1186/s12884-022-04945-z> | [https://www.thelancet.com/journals/eclinm/article/PIIS2589-5370\(22\)00123-](https://www.thelancet.com/journals/eclinm/article/PIIS2589-5370(22)00123-)

7/fulltext#seccesectitle0001). The authors should describe the mothers' vaccination status after the delivery, if the unvaccinated became vaccinated, etc.

We appreciate the potential bias introduced by maternal vaccination post-partum. We investigated the rates of post-partum vaccination during the study period and found that rates did not differ among the previously unvaccinated subgroups. **Change appears on p. 5 and in the reference list nos. 16-17.**

	Sub-group	Infants born to Naïve and Natural Immunity sub-group mothers (at delivery) in the study cohort (n)	Infants born to Naïve and Natural Immunity sub-group mothers (at delivery) vaccinated post-partum , during study period (n)	% Within subgroup
Cases	Naïve	228	22	9%
	Natural immunity	46	5	10%
	Naïve + Natural	274	27	9%
Controls	Naïve	14,699	1134	7%
	Natural immunity	5430	576	10%
	Naïve + Natural	20,129	1710	8%

Other points:

It is necessary for a better description of the survival analysis. How was the matching procedure accounted for in the Cox regression? As it is a case-control study, the matching strata can't be ignored.

Thank you for this comment. In this matched case-control study, data analysis employed stratified Cox regression, where each case and its controls account for one stratum, and time is taken as a constant. This method is similar to conditional logistic regression with variable number of controls per one case. In order to clarify this point, this was added to the Methods. **Change appears on page 16.**

Did the authors check for proportional hazards? In the presence of covariates and non-proportional hazards, the standard error is misestimated and needs to be estimated using Bootstrap. (<https://doi.org/10.1001/jama.2020.1267>)

As mentioned above, since time is a constant for all strata, and therefore practically ignored, there is no need to test for non-proportional hazards. In these stratified Cox analyses, time is taken as a constant and essentially ignored.

In the discussion Line 247. There is information about a 2.8-fold increase in hospitalisation without reference to a text or paper.

This is correct. During the Delta variant surge 173 infants were admitted, whereas 486 infants were admitted during the Omicron surge, a 2.8- fold increase. We have clarified in the results. **Change appears on p. 6.**

The length of stay is similar across all groups. Do the authors have any hypotheses about it? A paragraph about it should enrich the discussion.

Despite significant differences in infant hospitalization rates based on maternal immunity, we observed similar hospitalization durations across all groups, regardless of maternal immunization status. We postulate that the uniformity in stay length may be due to a shared clinical presentation, primarily fever in infants, which necessitates brief hospital stays. However, it is important to note that we lack data regarding the prevalence of fever. **Change appears on p. 10.**

Line 118-127. Length of stay tends to be very right-skewed, so a median (interquartile range) should be described better.

The median and interquartile range of length of stay are now reported in **Table 1 and in the Results, changes appear on p. 6).**

Line 153 (84%, 95% CI: 75-90% vs. 56%, 95% CI: 39-68%, $p < 0.001$) – Is this p-value from a contrast between maternal hybrid versus natural groups? **From all other groups: change appears on p. 6.**

A histogram showing the age, in days, of the hospitalisation of the cases is helpful. As the study period covers only 206 days, it should have a very low number of individuals, between 150-180 days.

We created a histogram showing the infant age at hospitalization among the cases, Suppl Figure 1. **(Change appears in Suppl figure 1 and on p. 4-5 of the Results.)**

Please change phrases using “recent” to the publication year so we can grasp the temporal association. (In discussion)

Done.

The discussion lacks a paragraph about the limitations of the study.

As mentioned in response to the earlier comment, description of the study limitations was added to the Discussion.

What test was applied to the Bonferroni correction? (Methods line 335), if not used, remove it from methods.

Bonferroni correction was used to investigate the difference between the immunity subgroups (Table 1). **Change appears in the Results, p. 6**

Methods: Line 291 states the study start is February 27, 2021; however, all other parts of the text state August 24.

We apologize if this was not clear. We have clarified the study period dates (Aug 24, 2021 – 15 March, 2022) and the limit of infant date of birth as a parameter for inclusion in the study group cases or controls, which was defined as infants up to six months of age during the study period. **Change appears in the Methods, p. 14)**

Supplementary Figure 2: It lacks the scale of the axis Y

Thank you for pointing this out, Y-axis ticks and scales were added to **supplementary figure 3A-B (formerly 2A-B) change appears on the figures and in the Results, p 5.**

Code availability: Please provide the code used in the analysis in an open repository (zenodo, GitHub, etc.)

Owing to the presence of personal information in our dataset, we regretfully cannot share the code and data currently. The protection of individual privacy holds highest priority for us, and we approach the task of securing this data with utmost seriousness.

We acknowledge the value of fostering open collaboration and sharing knowledge. Hence, upon obtaining the requisite approval from the Ministry of Health to align with data privacy regulations and standards, we would be happy to share both the code and data.

REVIEWER COMMENTS

Reviewer #1 (Remarks to the Author):

The authors have adequately addressed my concerns with their revisions. I defer to the statistical reviewer if the updated methods are appropriate. The main study findings are important despite new variants and updated immunization strategies.

Reviewer #2 (Remarks to the Author):

The authors have partially addressed the comments in my review. I am still not clear that the term estimated effectiveness can be used for a case-control design. I still think this paper should have a detailed review from a statistician prior to a decision being made.

Reviewer #3 (Remarks to the Author):

The manuscript is much improved. Thank you for the revision.

I have some comments. One that it may be late for, but I think is important is the decision to call two vaccine doses "partial." (See my final comment). I think that goes against most definitions (1 dose), despite Israel's wise decision to recommend 3rd doses for pregnant patients.

Line 125, 324-326: One comment on "for/with" and one (more important one) about the choice to exclude certain cases that were deemed not "for" Covid.

The lack of difference in PICU admission would suggest that reduced infections (not severity of each infection) is behind the overall findings. That is fine, but then I wonder why bother limiting the analysis to "For Covid" as opposed to "With Covid." Table S1 shows us the exclusions. I don't understand why authors excluded a lot of these. The top 2 in particular are odd to me. Infectious diseases would seem to imply >1 infection at once. Would it not follow that simultaneous SARS2 infection may have taken what otherwise might have been a mild illness and rendered the infant sick enough to warrant hospitalization? Why exclude this? And why exclude infants born SARS2+? Again, if the mother was vaccinated more than a few days prior, it would follow that these infants had antibodies. They should be included, in my view. They count. (That is, if hybrid immunity reduces the number of babies born SARS2+, we would very much like to know that).

Line 175-188 and Line 241: I remain somewhat confused about this, though I think I understand (which means: it needs to be more clearly explained to readers). It is stated that protection from immunity stimulation before pregnancy is greater than stimulation during the first 20 weeks of pregnancy (VE=86 vs VE=78). But that is the "overall situation," and not the case for vaccination. This is an example of "misleading averages." If one person is driving 100km/hr and another 10 km/hr, the average rate is 55 km/hr. But nobody is driving anywhere near that rate. So: Figure 3 shows that for Infection <20 weeks + Hybrid, VE=83. That is slightly higher than the VE for infection prior to pregnancy + hybrid (VE=79). For NATURAL, the VE is higher from infection before pregnancy compared to infection <20 weeks (VE=43 vs VE=59). So it seems to me that immunity from infection <20 weeks pregnancy is not as protective as infection before pregnancy, but this is not the case for vaccine-conferred immunity. If I understand this correctly, when you combine this all, you see that immunity gained prior to pregnancy looks greater than that during 0-19 weeks. But that apparently is driven by the weaker immunity to natural infection during early pregnancy, not weaker immunity in response to vaccination at that time. This is a major distinction. While I agree that 3rd trimester is probably optimal, the way it is presented, one might incorrectly conclude that additional vaccination from 0-19 weeks is not as good as pre-pregnancy.

That appears untrue. I'm also not clear how the authors conclusions on this are consistent with Figure S5 (Hybrid analysis)—or at least, I think they need to do a better job of delineating this from the statement in Line 241. It's quite situational and I think that bares noting and clearly pointing out.

Line 259-262: Smaller thing but I remain unconvinced that Omicron is actually milder than Delta. Omicron certainly evades our existing Wuhan antibodies better. And yes, in the Naïve group (Fig 2), the case hospitalization rate is lower for Omicron. But how much of that is triage (that is, by Omicron, clinicians may have felt that an infant with SARS2+ and a mild fever could go home, which may not have been the case during Delta; also when hospital capacity is tight, clinicians may send home infants who they would prefer to admit to the ward; Omicron had this effect in many places). Also, a lot of people do not know they had prior immunity. So, by the Omicron era, a higher proportion of mothers may have actually had infections (but not realized it). So this would mean that at least some "naïve" mothers in the Omicron era were not in fact naïve. Therefore, the lower hospitalization rate we see there reflects immunity, not a difference in the variant's virulence.

Line 349-354. I really have trouble with 2 doses being considered "partial." By that argument, someone who is 80 years old today is only partially vaccinated if they have received 7 or 8 doses? Then, in Figure 4, the bins have 2 or more vaccine as the "most vaccinated" category. So that's a bit inconsistent. I think two doses should *not be considered partial. (I do agree, however, that additional doses during pregnancy are warranted, and again, I applaud Israel's early recognition of that.)

Reviewer #4 (Remarks to the Author):

The authors have addressed most of my previous major concerns, and their responses to others are acceptable.

Only about the Cox regression, the stratified model allows different baseline hazards within each stratum. Time is not considered as constant, but a non-proportional hazard won't misestimate the standard variance because the model only includes the exposure variable plus the strata, so if it is a non-proportional HR, it can be interpreted as weighted HR during the follow-up.

I do not have any further comment.

We thank the editors and reviewers for the careful re-reading of our manuscript. We have attended to all comments, our responses appear below. Changes appear highlighted in the manuscript file.

REVIEWER COMMENTS

Reviewer #1 (Remarks to the Author):

The authors have adequately addressed my concerns with their revisions. I defer to the statistical reviewer if the updated methods are appropriate. The main study findings are important despite new variants and updated immunization strategies.

We thank the reviewer for their careful reading of our manuscript.

Reviewer #2 (Remarks to the Author):

The authors have partially addressed the comments in my review. I am still not clear that the term estimated effectiveness can be used for a case-control design. I still think this paper should have a detailed review from a statistician prior to a decision being made.

We thank the reviewer for their careful reading of our manuscript. We refer the reviewer to our earlier response – that this design has been employed previously in many studies, including on COVID-19.

Reviewer #3 (Remarks to the Author):

The manuscript is much improved. Thank you for the revision.

I have some comments. One that it may be late for, but I think is important is the decision to call two vaccine doses “partial.” (See my final comment). I think that goes against most definitions (1 dose), despite Israel’s wise decision to recommend 3rd doses for pregnant patients.

Thank you for your careful reading of the ms. In the interest of clarity, a note was added to Table 1 and the relevant figures, denoting Partial vaccination: 1-2 doses; Full vaccination: 3-4 doses.

Line 125, 324-326: One comment on "for/with" and one (more important one) about the choice to exclude certain cases that were deemed not “for” Covid.

The lack of difference in PICU admission would suggest that reduced infections (not severity of each infection) is behind the overall findings. That is fine, but then I wonder why bother limiting the analysis to “For Covid” as opposed to “With Covid.” Table S1 shows us the exclusions. I don’t understand why authors excluded a lot of these. The top 2 in particular are odd to me. Infectious diseases would seem to imply >1 infection at once. Would it not follow that simultaneous SARS2 infection may have taken what otherwise might have been a mild illness and rendered the infant sick enough to warrant hospitalization? Why exclude this? And why exclude infants born

SARS2+? Again, if the mother was vaccinated more than a few days prior, it would follow that these infants had antibodies. They should be included, in my view. They count. (That is, if hybrid immunity reduces the number of babies born SARS2+, we would very much like to know that).

A: Thank you for this comment. We re-ran our analyses including these cases that were considered non-eligible. The results were overall similar to our results of the study cohort. The new analysis appears alongside the main results, in Supplementary Table 2 and in the Results (lines 146-148).

Line 175-188 and Line 241: I remain somewhat confused about this, though I think I understand (which means: it needs to be more clearly explained to readers). It is stated that protection from immunity stimulation before pregnancy is greater than stimulation during the first 20 weeks of pregnancy (VE=86 vs VE=78). But that is the “overall situation,” and not the case for vaccination. This is an example of “misleading averages.” If one person is driving 100km/hr and another 10 km/hr, the average rate is 55 km/hr. But nobody is driving anywhere near that rate.

So: Figure 3 shows that for Infection <20 weeks + Hybrid, VE=83. That is slightly higher than the VE for infection prior to pregnancy + hybrid (VE=79). For NATURAL, the VE is higher from infection before pregnancy compared to infection <20 weeks (VE=43 vs VE=59). So it seems to me that immunity from infection <20 weeks pregnancy is not as protective as infection before pregnancy, but this is not the case for vaccine-conferred immunity. If I understand this correctly, when you combine this all, you see that immunity gained prior to pregnancy looks greater than that during 0-19 weeks. But that apparently is driven by the weaker immunity to natural infection during early pregnancy, not weaker immunity in response to vaccination at that time. This is a major distinction. While I agree that 3rd trimester is probably optimal, the way it is presented, one might incorrectly conclude that additional vaccination from 0-19 weeks is not as good as pre-pregnancy. That appears untrue. I’m also not clear how the authors conclusions on this are consistent with Figure S5 (Hybrid analysis)—or at least, I think they need to do a better job of delineating this from the statement in Line 241. It’s quite situational and I think that bares noting and clearly pointing out.

A: Thank you for the comment, it has highlighted two issues with the text that we have addressed. Firstly, the direction to Supplementary Figure 4 at this point should have read Supplementary Figure 5. This has been corrected. In light of the reviewer’s comments we consulted with the team and other experts. Since the observed differences were minor, and the physiological response is beyond the scope of the present manuscript, we decided to attenuate our remarks on this issue. The relevant speculative remark (and references) were deleted (248-249).

Line 259-262: Smaller thing but I remain unconvinced that Omicron is actually milder than Delta. Omicron certainly evades our existing Wuhan antibodies better. And yes, in the Naïve group (Fig 2), the case hospitalization rate is lower for Omicron. But how much of that is triage (that is, by Omicron, clinicians may have felt that an infant with SARS2+ and a mild fever could go home, which may not have been the case during Delta; also when hospital capacity is tight, clinicians may send home infants who they would prefer to admit to the ward; Omicron had this effect in many places). Also, a lot of people do not know they had prior immunity. So, by the Omicron era, a higher proportion of mothers may have actually had infections (but not realized it). So this

would mean that at least some “naïve” mothers in the Omicron era were not in fact naïve. Therefore, the lower hospitalization rate we see there reflects immunity, not a difference in the variant’s virulence.

A: This is an interesting point and we appreciate the Reviewer’s pointing this out. The rate of “naïve” in the Omicron wave may be an overestimation, so it follows that we have a difference in protection between the two groups, since many supposed “naïve” patients in fact had some immunity. However, we have no way of tracking asymptomatic (unrecorded) infections in our database. We have moderated our remarks in the Discussion at the relevant paragraph (258 and ff) and in the Limitations.

Line 349-354. I really have trouble with 2 doses being considered “partial.” By that argument, someone who is 80 years old today is only partially vaccinated if they have received 7 or 8 doses? Then, in Figure 4, the bins have 2 or more vaccine as the “most vaccinated” category. So that’s a bit inconsistent. I think two doses should *not be considered partial. (I do agree, however, that additional doses during pregnancy are warranted, and again, I applaud Israel’s early recognition of that.)

A: Figure 4 considers patients with “Natural” or “Hybrid” immunity. We compared to subgroups of Hybrid immunity patients: those who received only 1 vaccine dose and contracted COVID vs. those who received 2 or more doses and contracted COVID, compared to “Natural” immunity as reference group. This was in order to emphasize the added value of booster vaccine doses, even in patients who contracted COVID, similar to seasonal influenza vaccination.

Reviewer #4 (Remarks to the Author):

The authors have addressed most of my previous major concerns, and their responses to others are acceptable.

Only about the Cox regression, the stratified model allows different baseline hazards within each stratum. Time is not considered as constant, but a non-proportional hazard won’t misestimate the standard variance because the model only includes the exposure variable plus the strata, so if it is a non-proportional HR, it can be interpreted as weighted HR during the follow-up.

I do not have any further comment.

A: We thank the Reviewer for this comment, we now added the term “weighted HR” in the Data Analysis section